# 🦉 GPT4Video: A Unified Multimodal Large Language Model for Instruction-Followed Understanding and Safety-Aware Generation

## ABSTRACT

Recent advances in Multimodal Large Language Models (MLLMs) have constituted a significant leap forward in the field, particularly in the processing of videos, which encompasses inherent challenges such as spatiotemporal relationships. However, existing MLLMs are predominantly focused on the comprehension of video inputs, with limited capabilities in generating video content. In this paper, we present GPT4Video, a unified framework that seamlessly and lightly integrates with LLMs, visual feature extractors, and stable diffusion generative models for cohesive video understanding and generation. Moreover, we explore a `text-only finetuning` approach to equip models for instruction-following and safeguarding in multimodal conversations, enhancing training efficiency and generalization capabilities. Additionally, we construct multi-turn and caption-interleaved datasets for finetuning and benchmarking MLLMs, which serve as solid resources for advancing this field. Through quantitative and qualitative assessments, GPT4Video demonstrates the following advantages: 1) The framework incorporates video generation ability without adding extra training parameters, ensuring seamless compatibility with various video generators. 2) The model achieves superior performances across a variety of benchmarks. For instance, it outperforms Valley [28] by 11.8% on video question answering, and surpasses NExt-GPT [48] by 2.3% on text-to-video generation. 3) As safety pioneers in open-source MLLMs, we developed finetuning and evaluation datasets, securing an F1 score exceeding 80% in blocking harmful content during understanding and generating videos. In general, GPT4Video shows potential to function as a real-life assistant, marked by its effectiveness, adaptability, and safety. We will open-source our code, data, and models.

## CCS CONCEPTS

• **Computing methodologies**; • **Artificial intelligence**; • **Philosophical/theoretical foundations of artificial intelligence**; • **Cognitive science**;

## KEYWORDS

Multimodal Large Language Model, Video Understanding and Generation, Instruction-Following, Safeguarding, Data Construction.

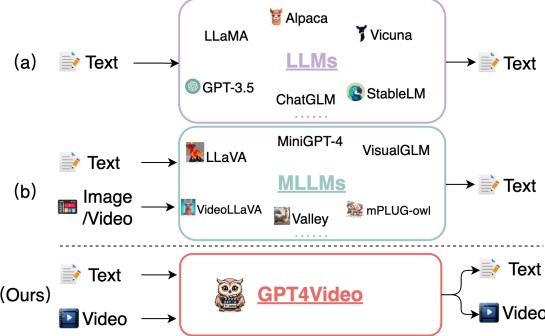

**Figure 1: Pipeline comparison of our GPT4Video with the current LLMs and MLLMs.**

## 1 INTRODUCTION

Large language models (LLMs) such as LLaMA [43], ChatGLM [13], Vicuna [11] (in Figure 1 (a)), have undergone significant advancements, paving the way for general artificial intelligence (AGI) and showcasing remarkable zero-shot capabilities across diverse linguistic tasks. Based on this foundation, multimodal large language models (MLLMs) like MiniGPT-4 [63], Video-LLaMA [55], Macaw-LLM [30] (in Figure 1 (b)) have been successively introduced. These models are capable not only of processing textual information but also of handling data across various modalities such as images, audio, and videos, demonstrating exceptional performance in multimodal tasks [54]. Nevertheless, current MLLMs primarily focus on processing multimodal inputs, yet they fall short in generating multimodal information on the output side. For a refined AGI system [17], it is crucial not only to understand multimodal inputs but also to efficiently generate them, mirroring human interaction in real-world settings.

Nonetheless, endowing LLMs with multimodal generation capabilities presents a significant challenge: *what* content aligns with user requirements, and *when* to generate it appropriately. Several recent studies have made preliminary forays in this direction [23, 48, 62]. GILL [23] initially investigated leveraging LLMs for image generation, introducing "generative vokens" (i.e. the special token "") into the text vocabulary of LLMs. NExt-GPT [48] further extends the concept of "generative vokens" to a variety of modalities (e.g. image, audio, and video). Though the approaches have demonstrated certain efficacy, they have several inherent limitations:

(1) The representational capacity of "generative vokens" is limited. Due to the constraints of their training mechanism, they fail to fully harness the outstanding potential of LLMs in text generation and emergence ability.

(2) Adding new vocabulary to the LLM's lexicon may disrupt the LLM's original capabilities.

(3) These methods lack the flexibility to adapt different generation models. For instance, upgrading the text-to-image/video model necessitates retraining the LLM.

Note that, quantitative analyses on the representative method [48] in Section 5 provide support for the above claims.

To address the limitations, we present *GPT4Video*, a unified framework for augmenting LLMs with both video understanding and generation capabilities (in Figure 1 (Ours)). Inspired by previous success [53], we leverage the robust capabilities of pretrained LLM and visual feature extractor for *video understanding*. Considering the the inherent complexity of video over images, we introduce a video abstractor with a dual attention mechanism [14, 61] to facilitate precise alignment between video and text. For *video generation*, instead of incorporating "generative vokens", we train LLMs to produce the video descriptions with tags at the appropriate moment. These tags and descriptions respectively act as trigger ("*when*") and input prompts ("*what*") for a pre-trained text-to-video model.

Previous studies commonly created and then finetuned MLLMs using multimodal instruction datasets for understanding abilities (e.g. text-video interleaved data) [8, 10, 30]. In contrast, we investigate a *text-only finetuning* method for video generation, replacing video with its caption in instruction datasets during the second-stage training.[1] Accordingly, we construct datasets and apply the proposed finetuning for two scenarios: equipping models for instruction-following (45K instances) and safeguarding (4K instances). Our GPT4Video has the following appealing advantages:

- **Seamless Integration**: The framwork requires no additional modules or training parameters, thus can be seamlessly integrated into a variety of text-to-video models (e.g. Zero-Scope [29] and VideoCrafter [9]).
- **Instruction-Followed Video Generation**: Textual data has more powerful abstraction and expression capabilities in summarizing content. The fine-tuning with our text-based instructions is expected to efficiently harness LLMs' generalization abilities in generating content-rich video prompts that align with instructions.
- **Safeguarding**: Through a simple safety alignment, our model substantially rejects both the processing of harmful video inputs and the generation of harmful video outputs. This makes it one of the first models to incorporate safeguarding features among open-source MLLMs.[2]

We conduct experiments on a variety of multimodal benchmarks, including open-ended question-answer [7, 50], video captioning [50], text-to-video generation [50] and safeguarding (ours). Results show that our model significantly and consistently outperforms the advanced models, NExt-GPT [48], demonstrating the effectiveness and universality of our framework and approach. Our **main contributions** are:

- We propose a unified framework that enhances LLMs with video understanding and generation capabilities through

---

[1]The assumption is that the model has acquired vision-language knowledge at the first-stage training [27, 63].

[2]We primarily focus on explicit content, as pornography represents the most significant risk in the realm of video generation.

seamless and lightweight integration with pre-trained Abstractor and T2V models, demonstrating remarkable extensibility.

- We explore a simple and effective method, text-only finetuning, for developing instruction-followed models, showing substantial improvements in performance. As safety pioneers, we further employ this approach for safety alignment, offering an alternative to einforcement learning from human feedback (RLHF).
- We build and release valuable datasets to facilitate future work on multimodal LLMs, including (1) a instruction dataset for video generation that covers a wide range of conversational scenarios; (2) a safety benchmark for evaluating and enhancing safeguarding capabilities in video-interleaved conversations.

## 2 RELATED WORK

**Multimodal Language Models** Numerous studies have previously developed multimodal language models that can handle visual inputs and text outputs, or vice versa, such as [6, 52]. With these advancements of LLMs, some researches have focused on learning a joint embedding space for multiple modalities, as demonstrated in [16, 36]. Others have combined pre-trained single-modality models to showcase impressive zero-shot capabilities [1, 24]. More recently, there has been a growing interest in enabling multi-modal LLMs to follow instructions, as shown in [12, 53, 63]. To facilitate research in this area, [51] introduced MultiInstruct, the first multi-modal instruction tuning benchmark dataset covering a wide range of tasks and categories. Additionally, [27] explored multi-modal instruction-tuning using machine-generated data, while [30] fine-tuned all model parameters to allow the textual LLM to process four modalities.

**Large Language Models** Large language models (LLMs) commonly refer to as Transformer-based language models with billions of parameters [44] and have revolutionized the research paradigm in natural language processing community [13, 43]. Furthermore, recent works have demonstrated that supervised fine-tuning, also known as instruction-tuning, can effectively improve the zero-shot performance of these LLMs [11, 42]. Zhao et al. [60] present a comprehensive survey on the research of LLMs.

**Text-to-Image/Video Generation** Text-to-image/video generation refers to the task of producing realistic images or videos based on natural language descriptions. One of the earliest approaches to this task was the use of conditional GANs [37]. Since then, various techniques have been developed to improve the quality of the generated images [33]. Compared to text-to-image generation, text-to-video generation is relative new and still remains challenging. Previous approaches have utilized techniques such as VAEs with recurrent attention [32] and expanding GANs from image to video generation [26]. Diffusion models have also been used to generate videos in recent works [2, 18, 40, 47].

**Responsible AI** As the AI systems become increasingly powerful, developing responsible AI have drawn significant scientific attention recently [22]. Various works have pointed out the safety risks of LLMs, such as toxicity [39], and hallucination [57]. The safety of LLMs is commonly measured by specialized benchmarks, such as

**Figure 2: Architectural Overview of GPT4Video.** This framework has two components for video processing. The video encoding module employs a frozen ViT-L/14 model to capture raw video features, while the video abstraction module utilizes a transformer-based cross attention layer and two novel learnable tokens, designed to condense video features. The core of GPT4Video is powered by a frozen LLaMA model, efficiently fine-tuned via LoRA. The LLM is trained with custom video-centric and safety-aligned data, enabling it to comprehend videos and generate appropriate video prompts (indicated by underlined text). These prompts are then used to create videos from the Text-to-Video Model Gallery. Icons of a snowflake and a flame visually distinguish between the non-trainable and trainable parameters within the system. The two bullet points highlight GPT4Video's dual capabilities in understanding and generating video content.

RealToxicityPrompts on toxicity [15]. More recently, [58] present SafetyBench, a large-scale diverse set of multiple choice questions across several aspects of safety concerns.

## 3 OUR FRAMEWORK

We introduce the GPT4Video, a unified framework designed to endow LLMs with advanced video understanding and generation proficiencies. As shown in Figure 2, the architecture is composed of three integral components: 1) *video understanding module* that employs a video feature extractor and a video abstractor to encode and align video information with the LLM's word embedding space; 2) *LLM*, utilizing the structure of LLaMA and employing parameter-efficient fine-tuning (PEFT) methods, specifically LoRA [19], while keeping the original pre-trained parameters intact; and 3) *video generation part* that conditions the LLM to generate prompts for a model from Text-to-Video Model Gallery through a meticulously constructed instruction dataset.

### 3.1 Video Understanding Module

**Visual Encoder** Given a video denoted as $\mathbf{v}$, we uniformly sample $T$ frames, which are represented as $\mathbf{v} = [\mathbf{v}_1, \mathbf{v}_2, ..., \mathbf{v}_T]$. For each individual frame $\mathbf{v}_i$ (where $i$ ranges from 1 to $T$), we utilize a pre-trained CLIP visual encoder to extract its visual features, denoted as $\mathbf{f}_v^i = \mathbf{ViT}(\mathbf{v}_i)$. Here, **ViT** represents the Vision Transformer used in CLIP for feature extraction. After extracting the visual features $\mathbf{f}_v^i$ from each frame, we compile the video features

$\mathbf{F}_v = [\mathbf{f}_v^1, \mathbf{f}_v^2, ..., \mathbf{f}_v^T]$. More specifically, GPT4Video adopts the CLIP ViT-L/14 model [36] for the visual encoding task. This model comprises 24 layers, with each layer having a hidden dimension of 1024 and processing patches of size 14. Within this module, images are uniformly resized to a standard dimension of 224 x 224 pixels and then divided into patches using a stride of 14 pixels. These patches are treated as input tokens for the transformer block, enabling its self-attention mechanisms to effectively generate detailed image embeddings. Consequently, the output of **ViT** contains 256 spatial patch features and 1 global feature (identified as the "[CLS] token"), resulting in the image features $\mathbf{f}_v^i \in \mathbb{R}^{257 \times 1024}$ and video features $\mathbf{F}_v \in \mathbb{R}^{(T*257) \times 1024}$.

**Video Abstractor** The input sequence for video features, denoted as $\mathbf{F}_v \in \mathbb{R}^{(T*257) \times 1024}$, becomes considerably large, particularly with an increased number of sampled frames, $T$. This results in a substantial computational challenge when processing through a LLM. To mitigate this, we have implemented a video abstractor. This tool efficiently condenses the visual information into a few learnable tokens, thereby generating high-level visual representations and significantly reducing the computational demands. Many image-based MLLMs [24, 53] achieve this by adopting a Q-former-like structure, which incorporates learnable query tokens to distill image features. Considering the complexity of video over image, we further introduce a learnable enhance tokens to augment video feature extraction by forming a dual attention mechanism.

We have validated the effectiveness of this structure in our experiments detailed in Table 5. We mathematically define query tokens as $\mathbf{Q}_s \in \mathbb{R}^{N_s \times D}$ and enhance tokens as $\mathbf{Q}_t \in \mathbb{R}^{N_t \times D}$, where $N_s$ and $N_t$ denote the counts of query and enhance tokens, respectively, and $D$ signifies the dimension of the token embeddings.

The $\mathbf{Q}_s$ and $\mathbf{Q}_t$ respectively compute cross attention with the video features $F_v$. Our cross attention module [44] consists of six encoder layers, wherein the pivotal component of each layer is the multi-head self-attention mechanism (MSA) [44].

We treat $\mathbf{Q}_s$ and $\mathbf{Q}_t$ as queries, respectively, and the result of concatenating them with the video feature $\mathbf{F}_v$ serves as the key and value to compute cross attention. This process yields video features $\mathbf{F}_s \in \mathbb{R}^{N_s \times D}$ and $\mathbf{F}_t \in \mathbb{R}^{N_t \times D}$, respectively. We set $N_s$ and $N_t$ to be the same and then sum the query and enhance features to obtain the final video feature $\hat{\mathbf{F}}_v \in \mathbb{R}^{N_s \times D}$. The mathematical expression is as follows:

$$\mathbf{F}_s = \text{CrossAttention}(\mathbf{Q}_s, [\mathbf{F}_v; \mathbf{Q}_s], [\mathbf{F}_v; \mathbf{Q}_s]) \tag{1}$$

$$\mathbf{F}_t = \text{CrossAttention}(\mathbf{Q}_t, [\mathbf{F}_v; \mathbf{Q}_t], [\mathbf{F}_v; \mathbf{Q}_t]) \tag{2}$$

$$\hat{\mathbf{F}}_v = \text{LN}(\mathbf{F}_s + \mathbf{F}_t) \tag{3}$$

where ";" indicates concatenation, $\text{LN}(\cdot)$ denotes layer normalization [3].

Finally, we employ a linear mapping layer to project the low-dimensional video feature $\hat{\mathbf{F}}_v$ into the high-dimensional word embedding space of the LLM, mathematically denoted as $\mathbf{F}_{video} = \mathbf{W}\hat{\mathbf{F}}_v + \mathbf{b}$, where $\mathbf{W}$ and $\mathbf{b}$ are the learnable weight matrix and bias vector, respectively, of the linear layer.

## 3.2 Video Generation Module

Existing model [48] of integrating "generative vokens" only accommodates one specific text-to-video model per training session. This limitation arises because different text-to-video models may utilize various text encoders. In contrast, GPT4Video replaces "generative vokens" by generating T2V models' textual prompts. As demonstrated in the Text-to-Video Gallery, our approach is compatible with the full spectrum of text-to-video models, including Make-A-Video [40], Tune-A-Video [47], Text2Video-Zero [20], VideoCrafter [9], ZeroScope [29]. More importantly, should more advanced models be integrated into the Text-to-Video gallery in the future, GPT4Video will seamlessly adapt without necessitating any modifications. In this work, we employ ZeroScope [29] [3] as our default video generation model. We also report VideoCrafter1 [9] in Table 6.

## 3.3 Large Language Model and Integration

LLMs have demonstrated remarkable capability in understanding and following human instructions. Thus, we leverage pretrained LLMs as our cognitive module. It is worth noting that the cognitive module also serves as the textual modality encoder in our approach. We adopt Llama-7B model with LoRA [19] as the LLM component of GPT4Video.

---

[3]https://huggingface.co/cerspense/zeroscope_v2_576w

## 4 OUR TRAINING METHOD AND DATASET

GPT4Video employs a two-stage training strategy. In the first phase, we focused on enabling GPT4Video to comprehend video content. Inspired by Video-ChatGPT's success with LLaVA's pretrained parameters, we used mPLUG-Qwl's parameters for GPT4Video's initial alignment. This laid a foundation in image comprehension. To tailor for video understanding, our video abstractor was further trained on VideoChat 11K, chosen for its 7K high-quality video-text alignments. Results show that built on the model's initial image understanding, a modest amount of high-quality data is sufficient to endow GPT4Video with robust video comprehension capabilities. In the second phase, we employed text-only finetuning to equip the model with capabilities for video generation and safety guarding. The training pipeline for GPT4Video and examples of the training data for stage two are illustrated in Figure 3.

## 4.1 Text-Only Finetuning

GPT4Video employs a two-stage training strategy. In the first phase, we freeze the parameters of LLM and focus on training the video extractor to effectively align video features with the LLM's feature space, to enhance the model's comprehension of video content. The second stage involves pure text-based instruction tuning, where we employ the LoRA method for efficient fine-tuning of the LLM, focusing solely on training the newly added parameters in the LLM. The input prompts for LLM are as follows:

```
###Human:<video>video_embed</video>
###Human:video_instruction
###AI:
```

where "video_embed" represents the video features $\mathbf{F}_{video}$, and it was only employed in the first stage. "video_instruction" refers to the instruction data constructed for videos.

During the first training phase, we utilized the VideoChat-11k dataset [25], which includes 7,000 detailed descriptions and 4,000 multi-turn conversations. In the second phase, we employed our self-constructed GPT4Video-50k dataset along with a safety-aligned dataset, which will be elaborated upon in Sec. 3.2 and Sec. 4.3.

We perform instruction-tuning of the LLM only on the text tokens, using its original auto-regressive training objective. Specifically, for a video instruction $\mathbf{X}_t$ of length $L$, conditioned on visual information $\mathbf{F}_{video}$, our loss function, captured as the negative log likelihood, is formulated:

$$\mathcal{L}(\theta; \mathbf{X}_t, \mathbf{F}_{video}) = -\sum_{i=1}^{L} \log p_\theta(x_i | \mathbf{F}_{video}, \mathbf{X}_{t,<i}), \tag{4}$$

where $\theta$ is the trainable parameters, $\mathbf{X}_{t,<i}$ is the text tokens before the current prediction token $x_i$.

## 4.2 Instruction-Following Dataset

Our method initially trains the LLM to learn how to generate textual prompts based on contextual information, which in turn drives a pre-trained text-to-video model to produce videos. To effectively train the LLM to generate these prompts, we meticulously developed an instruction-following dataset named GPT4Video-50k. The fundamental idea behind constructing this dataset is to utilize video

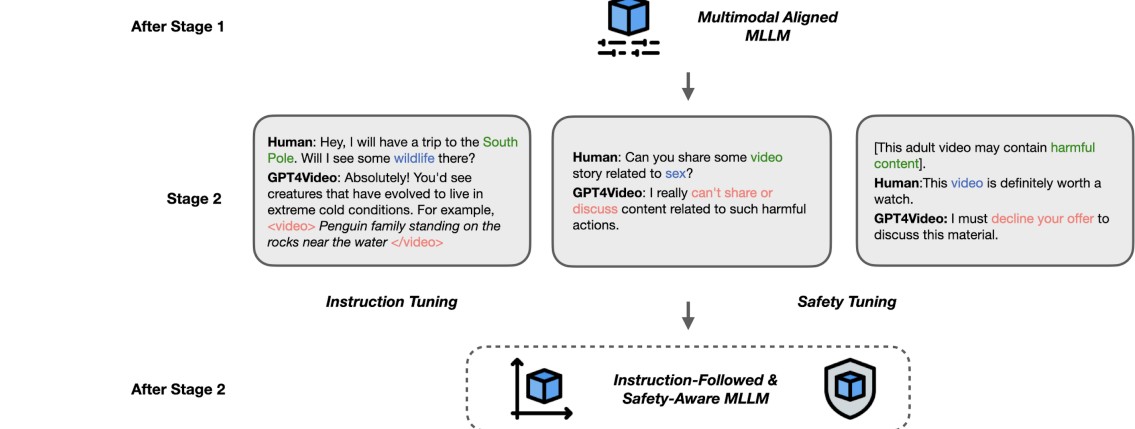

**Figure 3: The text-only finetuning pipeline and datasets. The left panel presents a data example for instruction tuning, while the center and right panels show data examples for safety tuning. These latter two panels respectively illustrate the model's safety measures at the output and input ends.**

descriptions in place of actual video content. Subsequently, we leverage the most advanced language model currently available, GPT-4, to create extensive dialogue data centered around these video descriptions.

More specifically, first, we randomly extracted 10,000 data examples from the Webvid10M [4] dataset and obtained their video descriptions. We used the string "<video> Video Caption </video>" as a placeholder for the actual video in the prompt to GPT-4, where "Video Caption" served as the placeholder for the video descriptions, and the "<video>" and "</video>" tags marked the boundaries of the video description. In the prompt, we require GPT-4 to construct three dialogue between two individuals (not a person and an assistant like most other LLMs) centered around the provided Video Caption (representing a real video). This approach makes our model's responses more reflective of real human emotions, rather than just those of an assistant. The complete example of dialogues and the detailed prompt, please refer to our supplementary materials.

Furthering our efforts, to endow the model with the capabilities for multi-turn dialogues and cross-video conversations, we developed a multi-round, interleaved instruction dataset centered around multiple video contents. Unlike the single-video centric dialogue data mentioned earlier, in this setup, we tasked GPT-4 with constructing rich conversational scenarios around the contents of multiple videos. Similarly, we utilized the format "<videoX> Video Caption </videoX>" to denote the actual videos, where "X" represents the video index, "Video Caption" is the description of the content of video X, and "<videoX>" and "</videoX>" serve as delimiters for the description's beginning and end. Specifically, we randomly selected 5000 samples from the WebVid10M [4] dataset and obtained their video descriptions. Diverging from the single-video centric data construction approach, our multi-video centric method requires at least two or more video descriptions as input prompts for GPT-4. To ensure that the dialogue data constructed by GPT-4 closely mirrors real scenarios and maintains logical coherence, we posit that these video descriptions should be semantically

related. To acquire pairs of semantically related video descriptions, we employed a text retrieval-based approach. The process involved three steps: firstly, using sentence transformers, we extracted feature embeddings for the video descriptions of the selected 5000 samples and the remaining descriptions in the WebVid10M dataset; secondly, we identified the top 10 similar video descriptions for these 5000 samples by calculating the cosine similarity between their feature embeddings; and finally, we randomly selected one or two from these top 10 descriptions to form pairs of video descriptions. For examples of the multi-video centric dialogues and the detailed prompts used for GPT-4, please refer to the supplementary materials accompanying this paper.

## 4.3 Safeguarding Dataset

For MLLMs, ensuring the generation of both useful and safe content, especially visuals, is imperative for their broad adoption. Take the handling of sexually explicit content as an example: Current text-to-image models employ an ancillary NSFW (Not Safe For Work) detection system. This system, upon recognizing potentially inappropriate content, replaces the intended output with a simple black image. While this approach effectively blocks unsafe content, it can lead to unnecessary consumption of time and computational resources. Moreover, when used in LLM-based models, this approach may lead to a disconnection between generated text and visual content. To address this issue, we have developed a safety-aligned instruction dataset that trains the model to politely refuse to respond to inappropriate user requests, thereby eliminating the need to initiate the generation model.

Specifically, we utilized the real-toxicity-prompts dataset [15], which comprises 100k texts along with their corresponding Toxicity scores. This dataset includes various categories for detection such as sexually_explicit, identity_attack, flirtation, threat, insult, and severe_toxicity. Focusing on the sexually_explicit and severe_toxicity categories, we extracted 1,500 texts from each, selecting those with toxicity scores exceeding 0.9. We then tasked GPT-4 to construct

**Table 1: Zero-shot video Question Answering result on MSVD-QA, MSRVTT-QA datasets.**

| Model | MSVD-QA | | MSRVTT-QA | |
|---|---|---|---|---|
| | Accuracy | Score | Accuracy | Score |
| FrozenBiLM [52] | 32.2 | – | 16.8 | – |
| VideoChat [25] | 56.3 | 2.8 | 45.0 | 2.5 |
| LLaMA Adapter [56] | 54.9 | 3.1 | 43.8 | 2.7 |
| Video LLaMA [55] | 51.6 | 2.5 | 29.6 | 1.8 |
| Video-ChatGPT [31] | 64.9 | 3.3 | 49.3 | 2.8 |
| Valley [28] | 65.4 | 3.4 | 45.7 | 2.5 |
| GPT4Video (Ours) | **66.3** | **3.6** | **49.8** | **3.0** |

dialogues based on these texts, aiming to generate polite refusals as responses to such content.

In addition to ensuring the safety of generated content, we have also implemented safety measures at the input stage. To this end, we introduced an additional detection model specifically designed to assess the safety of user input. In cases where inappropriate content is input by users, we employ a method similar to the one described above, training the model to tactfully decline responding to such content (see data example in the righy panel of Figure 3). In the experimental section 5.3, we conducted an in-depth and detailed evaluation of our model's safety features. For examples of more Safeguarding dataset and the detailed prompts used for GPT-4, please refer to the supplementary materials accompanying this paper.

### 4.4 Safeguarding Benchmark

To investigate the safety of MLLMs in video understanding and generation, we construct a multimodel benchmark for not safe for work (NSFW) content. The benchmark examines the percentage of harmful content that is rejected by the MLLMs when presented with various video inputs and queried with harmful and common questions. Referring to a text-only NSFW benchmark [21], we collect a diverse collection of video clips and queries with varying levels of harmfulness and safety. In specific, our benchmark includes two classification tasks focused on video understanding and generation, respectively. Each task includes 60 queries with an equal distribution of harmful and safe content. In the video generation task, in addition to the three-level harmful contents [21], the queries include natural scenes, human and animal activities, and real and comic episodes to ensure a diverse range of topics. Additionally, we design various video and question pairs to prompt MLLMs to provide inappropriate responses in the video understanding task. In this way, our benchmark can provide a comprehensive evaluation of the safety capabilities of MLLMs in video understanding and generation.

## 5 EXPERIMENTS

### 5.1 Experimental Settings

**Datasets and Evaluation Metrics.** To ensure a fair comparison, we followed work [28, 31] and assessed the model's understanding abilities through the Zero-shot Video Question Answering task. We conducted a comprehensive quantitative assessment using two widely-accepted open-ended question-answer datasets: MSRVTT-QA [50], MSVD-QA [7]. These evaluations were conducted in a zero-shot setting, utilizing GPT-assisted assessments to gauge the model's performance. This process aims to quantify the accuracy of the model's predictions, assigning scores from 1 to 5. We further conducted video captioning task on MSRVTT [50] dataset, using BLEU-4 [34] and METEOR [5] as evaluation metrics.

On the other hand, in terms of evaluating video generation capabilities, we followed NExt-GPT [48] and conducted Text-to-Video Generation tasks on the MSRVTT [50] dataset, using Fréchet inception distance (FID) [35] and CLIPSIM [46] (average CLIP similarity between video frames and text) as evaluation metrics.

**Implementation Details.** We leveraged the pre-trained parameters of mPLUG-Owl [53] to enhance our model's capability in video understanding. We uniformly sample four frames from the video and set the number of temporal and spatial tokens to 64. In the first phase, we set the learning rate to 1e-4 and trained GPT4Video for two epochs. In the second phase, we adjusted the learning rate to 2e-5 and continued fine-tuning for an additional three epochs. All the experiments are conducted on 8×A100 40G GPUs.

### 5.2 Instruction-Following Results

**Zero-shot Video Question Answering.** To assess the comprehension capabilities of GPT4Video, following VideoChat [25], we conducted comparative experiments on two most widely-used video question answering benchmarks. The experimental results of zero-shot inference are presented in Table 1. Specifically, we compare GPT4Video with six State-of-the-art models, including Frozen-BiLM [52], VideoChat [25], LLaMA Adapter [56], Video LLaMA [55], Video-ChatGPT [31] and Valley [28]. As can be observed, our proposed model GPT4Video reaches SOTA on all four benchmarks.

**Zero-shot Video Captioning.** We further compared the model's performance on the video captioning task using the MSR-VTT dataset. We evaluated two traditional transformer-based methods, ORG-TRL [59] and GIT [45], along with three LLM-based approaches, including mPLUG-2 [49], CoDi [41], and NExT-GPT [48]. As shown in Table 2, our model achieved better performance on both the BLEU-4 and METEOR metrics.

**Zero-shot Text-to-Video Generation.** We compare GPT4Video on the MSR-VTT [50] dataset for text-to-video generation task in a zero-shot manner quantitatively in Table 3. Following prior works [2, 48], we assess the visual quality using the FID metric and evaluate semantic consistency through the average CLIP similarity between video frames and the corresponding text. We benchmarked GPT4Video against four text-to-video specialized models: CogVideo [18], MakeVideo [40], Latent-VDM [38], and Latent-Shift [2], as well as two LLM-based methods, including CoDi [41] and NExt-GPT [48]. The results, presented in Table 3, demonstrate that GPT4Video outperforms the aforementioned models across all evaluated metrics. It is noteworthy that while NExt-GPT integrates ZeroScope as the video decoder in its video generation process,

**Table 2: Video-to-text generation (video captioning) results on MSR-VTT [50].**

| Method | BLEU-4 (↑) | METEOR (↑) |
|---|---|---|
| ORG-TRL [59] | 0.436 | 0.288 |
| GIT [45] | 0.548 | 0.331 |
| mPLUG-2 [49] | 0.578 | 0.349 |
| CoDi [41] | 0.521 | 0.325 |
| NExT-GPT [48] | 0.584 | 0.385 |
| GPT4Video (Ours) | **0.587** | **0.391** |

**Table 3: Text-to-video generation results (zero-shot) on MSR-VTT [50].**

| Method | FID (↓) | CLIPSIM (↑) |
|---|---|---|
| CogVideo [18] | 23.59 | 0.2631 |
| MakeVideo [40] | 13.17 | 0.3049 |
| Latent-VDM [38] | 14.25 | 0.2756 |
| Latent-Shift [2] | 15.23 | 0.2773 |
| CoDi [41] | - | 0.2890 |
| NExT-GPT [48] | 13.04 | 0.3085 |
| GPT4Video (Ours) | **12.91** | **0.3194** |

GPT4Video demonstrates enhanced precision in generating video content. This observation reinforces the claims we made in the introduction 1. For qualitative comparison results, please refer to Figure 5.

## 5.3 Safeguarding Results

**Table 4: Performance comparison of safety Evaluation. "VU" and "VG" represent Video Understanding and Video Generation.**

| Task | Method | Acc. | Prec. | Rec. | F1 |
|---|---|---|---|---|---|
| VU | Video LLaMA [55] | 0.51 | 0.13 | 0.57 | 0.22 |
| | Video-Chat [25] | 0.50 | 0.06 | 0.50 | 0.12 |
| | NExT-GPT [48] | 0.55 | 0.13 | 0.80 | 0.23 |
| | **GPT4Video** (Ours) | **0.82** | **0.83** | **0.81** | **0.82** |
| VG | NExT-GPT [48] | 0.65 | 0.37 | **0.85** | 0.51 |
| | **GPT4Video** (Ours) | **0.85** | **0.90** | 0.82 | **0.86** |

In Table 4, we test the safety performance of recent MLLMs in terms of popular classification metrics on our constructed safety benchmark. The understanding (VU) results show that most MLLMs had a low precision score, below 0.15, when it came to rejecting harmful queries (refusal was treated as a positive in our experiment).

**Table 5: Comparison of Abstractors for V2T tasks on MSR-VTT.**

| | Abstractor mode | BLEU-4 | METEOR |
|---|---|---|---|
| #0 | GPT4Video (Ours) | **0.587** | **0.391** |
| #1 | Ours\Temporal token | 0.568 | 0.358 |
| #2 | Q-former-64 | 0.572 | 0.369 |
| #3 | Q-former-128 | 0.583 | 0.389 |
| #4 | S&T Pooling | 0.581 | 0.388 |

**Table 6: Text-to-video generation on MSR-VTT.**

| Method | T2V Model | FID (↓) | CLIPSIM (↑) |
|---|---|---|---|
| NExT-GPT | ZeroScope | 13.04 | 0.3085 |
| GPT4Video | ZeroScope | 12.91 | 0.3194 |
| GPT4Video | VideoCrafter1 | **12.73** | **0.3263** |

Differently, our method demonstrates a strong ability to reject harmful queries, while maintaining a high recall rate for general queries. Upon examining the failure cases, we find that current MLLMs can only refuse to respond to harmful queries with textual hints, such as 'this is a harmful video?' It suggests that these MLLMs' ability to detect harmful content is mainly derived from LLMs. In the generation tasks with text-only queries, our method achieves a high Precision of 90%, compared to NExT-GPT with only 37%. Our method takes a more conservative approach to generating content, which may result in the refusal to generate some general queries related to harmful content, such as sexual dancing. Despite this, our method still achieves a slightly lower but comparable recall Recall compared to NExT-GPT.

## 5.4 Analysis

**Qualitative Analysis.** In this section, we present some cases to demonstrate the superior video understanding and generation capabilities of GPT4Video, as shown in Figure 4, 5. More cases are provided in supplementary materials.

**Ablation Study.** In our ablation study presented in Table 5, we evaluated various configurations of the video abstractor. Experiment #1 omitted temporal tokens; #2 and #3 used Q-former with 64 and 128 query tokens, respectively; and #4 applied spatial and temporal pooling, a method used in Video-ChatGPT [31] and Valley [28]. Results on the V2T task clearly demonstrate that our GPT4Video's video abstractor achieves superior performance, validating its efficacy.

**Comparison of T2V models.** We initially selected ZeroScope model for evaluation due to its use in Next-GPT, ensuring a fair comparison. We have now incorporated evaluations using a more advanced VideoCrafter1 model. The results, as detailed in Table 6, demonstrate superior performance of our GPT4Video in text-to-video tasks. This enhancement underscores our method's capacity for seamless T2V model upgrades without necessitating modifications to the underlying LLM, unlike Next-GPT which requires complete retraining.

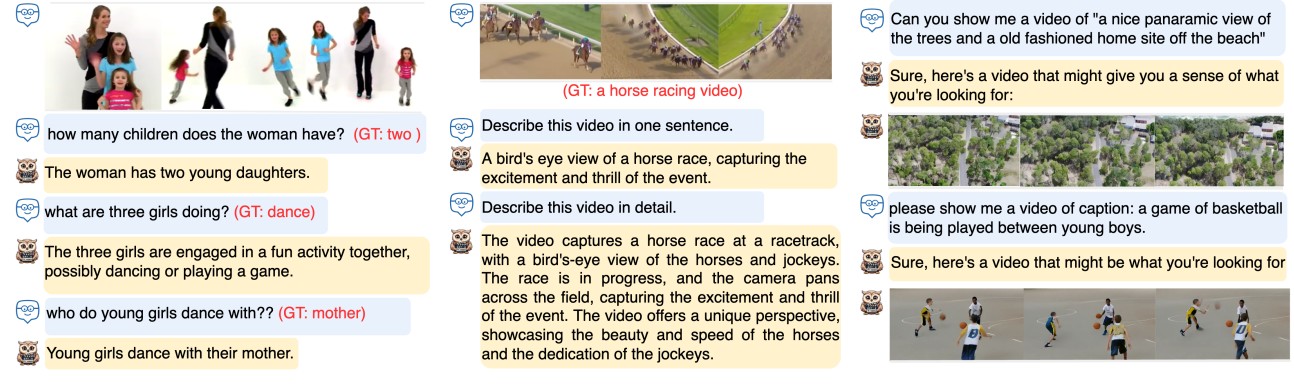

(a) Video Question Answering

(b) Video Captioning

(c) Text to Video Generation

**Figure 4: Visual examples of GPT4Video's responses on three tasks. The text in red represents the Ground-Truth.**

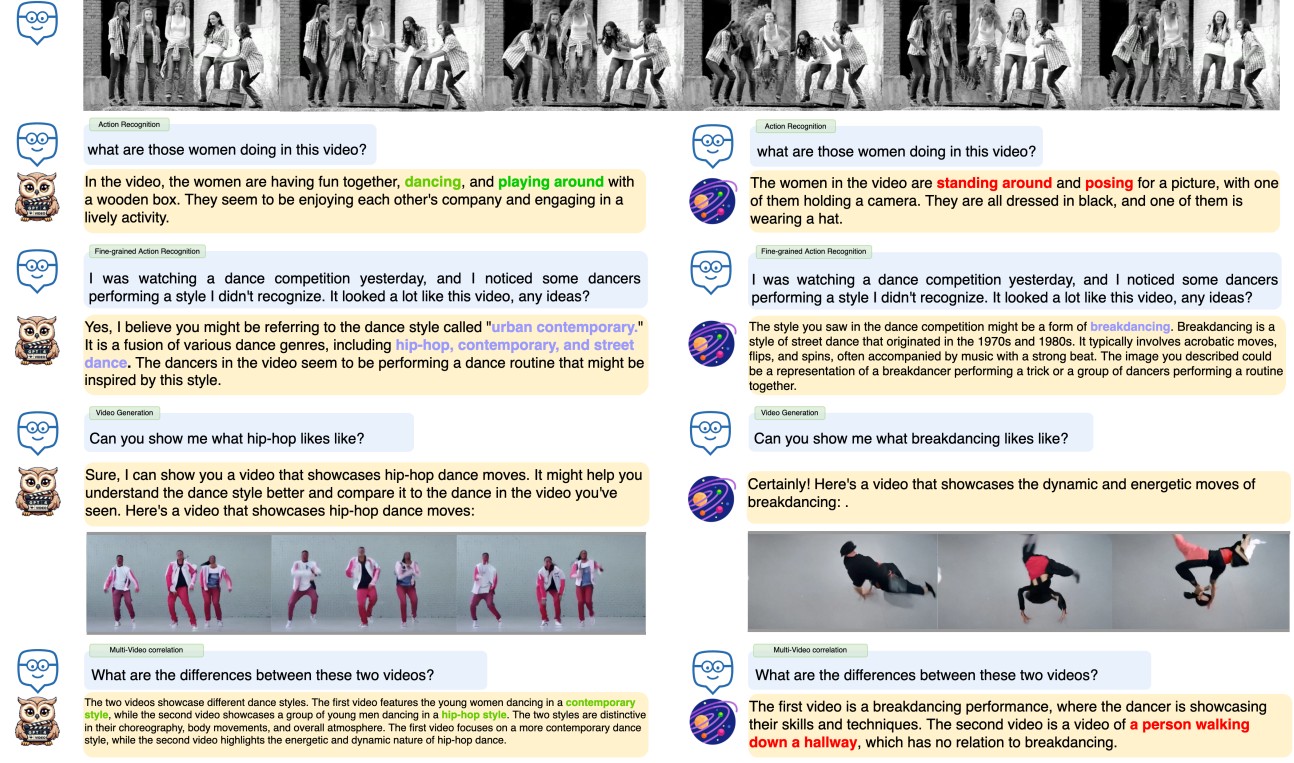

(Ours) GPT4Video

(b) NExt-GPT

**Figure 5: A demonstration comparing GPT4Video and NExt-GPT in multi-turn and interleaved conversation. The input video is about young women dancing tap. We highlighted key information using different colors to facilitate presentation.**

## 6 CONCLUSIONS

We present GPT4Video, a novel framework that significantly enhances Large Language Models with advanced video understanding and generative functions. Our approach leverages the descriptive power of LLMs to create detailed prompts for generative models, maintaining model simplicity and flexibility. The framework's effectiveness is underscored by its superior performance on multimodal benchmarks and its innovative approach to addressing content safety issues. The release of the specialized multi-modal instruction dataset promises to catalyze future research in the field. **Limitation:** GPT4Video currently specializes in video modality, with plans to expand to more modalities like image and audio in future updates. Additionally, while we have initiated steps to address content safety, our consideration of safety aspects is not yet exhaustive. In future work, we will strive to further refine and enhance this aspect.

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
