# OpenReview forum: "GPT4Video: A Unified Multimodal Large Language Model for lnstruction-Followed Understanding and Safety-Aware Generation"
_acmmm.org/ACMMM/2024/Conference — MM2024 Oral_

### Official Review · Reviewer_mq5A · 2024-05-24

**Rating:** 3
**Confidence:** 4

**Summary:**

This paper is about how to connect MLLMs and some video generation base models as tools for instruction-followed understanding and safety-aware generation.

**Strengths:**

1. construct multiturn and caption-interleaved datasets for finetuning and benchmarking MLLMs is a good contribution.
2. safeguarding is one of the important thing in text-to-video generation and needed to be considered.
3. this framework can do both text-to-video generation and video-to-text understanding.
4. using video abstractor to extract the features from video is good.

**Limitations:**

1. Although the dataset contribution, the method part seems to me is so engineering.
2. Since the author claim the flexible framework contributio, it would be good to see the more about the combination of different mllm and video generation models. (see only two models ZeroScope and VideoCrafter1, but no MLLM ablations found)
3. For caption task, usually we use CIDEr as the main metric, could the author also show that?

**Suitability:**

3

---

### Official Review · Reviewer_ikvB · 2024-05-28

**Rating:** 4
**Confidence:** 2

**Summary:**

1. GPT4Video is a framework that integrates video understanding and generation, with a focus on video understanding. It utilizes the parameters of mPLUG-Owl for fine-tuning, resulting in improved performance of video LLMs.

2. By conceptually integrating understanding and generation at the text level, existing text-to-video methods are integrated and invoked as repositories, achieving a fusion of understanding and generation.

3. A video QA annotation engine is obtained using the GPT4 multimodal model, which is centered around both single and multiple videos.

4. Regarding video safety, a benchmark for evaluation is proposed, and safety considerations are taken into account during fine-tuning.

**Strengths:**

1. Conceptually integrates video understanding and generation, expanding the capabilities and applications of large video models.
2. Proposes a video text annotation engine centered around multiple videos.
3. Introduces the concept of safety and creates a benchmark for safety.

**Limitations:**

1. The integration of text-to-video is simply done by using existing text-to-video methods as a library, lacking an ablation study on this module.
2. It may be worth considering merging understanding and generation at the feature level, which could be attempted as an ablation study.
3. In the data engine, could GPT4 be replaced with GPT-4V or a deployable vision-LLM?
4. Can comparisons be made with the latest video LLM for effectiveness?

**Suitability:**

2

---

### Official Review · Reviewer_bwuk · 2024-05-31

**Rating:** 4
**Confidence:** 2

**Summary:**

This paper introduces a text-only fine-tuning approach named GPT4Video that aims to enhance the video understanding and generation capabilities of LLMs. The authors propose using ViT to build a video abstractor and fine-tune Llama2 with video feature embeddings using LoRA. Experimental results demonstrate that GPT4Video outperforms existing models such as Video LLaMA, Video-Chat, and NExT-GPT in both video understanding and generation tasks.

**Strengths:**

1) This work is open-source, and the authors also provide their datasets to benefit the community.
2) The paper includes responsible AI considerations and proposes a safeguard dataset to address these concerns.
3) The paper is easy to follow, and the figures are well-drawn.

**Limitations:**

1) As one of the major contributions of this work, the necessity of the cross-attention mechanism needs more emphasis. Corresponding ablation studies can effectively demonstrate its efficiency. Currently, it is unclear why GPT4Video requires cross-attention and how the enhanced tokens and query tokens are generated (specifically, how Q, K, and V are obtained in Figure 2).

2) The authors claim in line 424 that they train the video extractor in the first phase. This is confusing—what does "video extractor" mean in this context? Is it the same as the Abstractor or the ViT? Please proofread the content to ensure coherence.

3) There are some typos in this manuscript, for example:
Line 182: "einforcement learning from human" should be "reinforcement learning from human"
Line 411: "we used mPLUG-Qwl’s parameters" should be "we used mPLUG-Owl’s parameters"

**Suitability:**

3

---

### Official Review · Reviewer_mwsP · 2024-06-03

**Rating:** 6
**Confidence:** 3

**Summary:**

This paper introduces GPT4Video, a framework that enhances Large Language Models (LLMs) with advanced video understanding and generative functions while also addressing content safety issues. GPT4Video employs a two-stage training strategy, with the first phase focusing on enabling the model to comprehend video content and the second phase focusing on text-only finetuning for video generation and safety guarding. The framework achieves state-of-the-art performance in zero-shot video question answering, video captioning, and text-to-video generation tasks on widely-accepted benchmarks. GPT4Video is a unified framework that integrates LLMs, visual feature extractors, and generative models for video understanding and generation. The results demonstrate the effectiveness of GPT4Video in bridging the gap between language and video understanding, showcasing its potential in various multimodal applications.

**Strengths:**

The paper demonstrates several strengths in terms of :
1. novelty:  It introduces GPT4Video, a framework that enhances Large Language Models (LLMs) with advanced video understanding and generative functions with state-of-the-art performance in video question answering, video captioning, text-to-video generation tasks and safeguarding against harmful content or generation.
2. methodology: The paper presents a two-stage training strategy for GPT4Video, focusing on video comprehension and text-only finetuning for video generation and safety guarding. The integration of LLMs, visual feature extractors, and generative models showcases a well-designed theoretical approach. Finally, the validation for the efficacy of their approach is backed by the ablation study in Table 5.
3. evaluation:  The paper provides comprehensive evaluations on various multimodal tasks. The comparisons with state-of-the-art models and the use of multiple evaluation metrics strengthen the evaluation process.
4. Clarity: The paper is very well-written and organized, making it easy to understand the proposed framework, experimental setup, and results.
5. Applications: The potential application of GPT4Video is significant in the world of multimodal understanding and in bridging the gap between language and video understanding

**Limitations:**

I think the paper is very well-written and detailed backed with substantial experiments. However, I would like to know if authors have performed any latency or resource utilization - comparison experiments on the tasks.

**Suitability:**

3

---

### Meta-Review · Area_Chair_Tjd7 · 2024-07-02

**Recommendation:** Accept (Oral)
**Confidence:** 5

**Metareview:**

The paper introduces GPT4Video, a video language model framework for video understanding and generation. The approach innovatively utilizes a text only finetuning strategy to achieve the dual aims of understanding and generation. In a departure from previous works on video and language alignment, it introduces mechanisms to compress the video content for suitable and efficient alignment. Another valuable contribution is the introduction of a safety-aligned dataset. The proposed approach outperforms prevailing models in terms of standard metrics for video understanding and generation.

The reviewers were mostly positive about the work. There were some concerns about the flexibility of the data engine in accommodating video models and minor concerns about grammar/typos. But overall, the proposed framework and datasets should serve as a good contribution for the multimedia and specifically, video understanding/generation community. **I recommend acceptance of this work.** As promised, the authors should open-source the materials related to this work.